# Protecting Human Health from Climate Change: Legal Obligations and Avenues of Redress under International Law

**DOI:** 10.3390/ijerph19095386

**Published:** 2022-04-28

**Authors:** Margaretha Wewerinke-Singh, Curtis Doebbler

**Affiliations:** 1Grotius Centre for International Legal Studies, Leiden University, 2311 ES Leiden, The Netherlands; 2Pacific Centre for Environment and Sustainable Development, The University of the South Pacific, Suva 0101, Fiji; 3Department of Law, University of Makeni, Makeni 10000, Sierra Leone; cdoebbler@unimak.edu.sl

**Keywords:** climate change, public health, human rights, redress, right to health

## Abstract

In this contribution, we explore how human health can be protected from climate change and its adverse effects by reliance on States’ obligations under international law. We achieved this by reviewing the principal legal instruments that establish the right to health, as well as those that recognize that climate change has an adverse impact on health (Part II). We then examine the means of redress that may be available to those whose human right to health has been interfered with or violated because of climate change (Part III). Finally, we draw some conclusions as to the current effectiveness and future direction of these developments.

## 1. Introduction

Climate change has been characterized as one of the defining challenges of our time [1]. The Nobel-Prize-winning Intergovernmental Panel on Climate Change (IPCC) has stated unequivocally in numerous reports that greenhouse gas-emitting human activities are causing global warming and associated damage to natural and human systems [2]. The Earth has now warmed by about 1.2 °C since pre-industrialization, and this poses increasingly severe threats to human health [3].

In this contribution, we explore how human health can be protected from climate change and its adverse effects by reliance on States’ obligations under international law. We achieve this by reviewing the principal legal instruments that establish the right to health, as well as those that recognize that climate change has an adverse impact on health (Part II). We then examine the means of redress that may be available to those whose human right to health has been interfered with or violated because of climate change (Part III). Finally, we draw some conclusions as to the current effectiveness and future direction of these developments.

Our contribution draws on the policy-oriented approach to international law and its general framework of enquiry into community values expressed primarily through the legal undertakings of States. We recognize that international law is part of a process whereby States strive to achieve values postulated in the international instruments [4]. We recognize the normative value of international law as both evidence of the value choices of States and as commitments by States that empower individuals and groups to make claims on States. Insofar as some States’ understanding of international law may differ from that which we suggest, we seek to rely on the expressions of States in their international instruments and understand them to have been made in good faith.

## 2. The Right to Health and the Climate Crisis

International climate change law and international human rights law both contain several widely ratified treaties that provide legal obligations for States in relation to both climate change and the right to health.

The relationship between climate change and health is recognized expressly in the United Nations Framework Convention on Climate Change (UNFCCC) [5]. This treaty refers to health in the first paragraph of its first article, stating that the adverse effects of climate change include its deleterious effects on “human health”, among other things [5]. The right to health is also mentioned specifically in the 2015 Paris Agreement [6]. Furthermore, in their national communications under the UNFCCC, States have recognized that climate change can adversely affect human health and that the state of the environment has a direct impact on human wellbeing [7]. For example, Samoa noted in its National Communication that non-physical health problems, i.e., psychological or emotional stress, “can frequently result from extreme weather events, particularly in instances where there is bereavement and damage to property and livelihood [8]”. Similarly, in their Nationally Determined Contributions (NDCs) under the Paris Agreement, about half of the States indicated that the physical and/or mental health of their people would be adversely impacted by climate change [9].

The relationship between the right to health and climate change has received increasing attention in recent years. Both in literature and in practice, it is recognized that climate change affects the right to health [10,11]. These impacts “stem from the immediate physical effects of climate change and the more gradual effects on the environment, human systems and infrastructure [12]”. The impacts of climate change on the right to health may occur “through both its direct impact and its impact on social support systems and cultural traditions [12]”. Moreover, these impacts may occur due to the adverse effects of climate change, as well as the adverse effect of ill-conceived measures to combat climate change or adapt to its effects [3]. 

In December 2019, five UN human rights treaty bodies [13] issued a Joint Statement on the IPCC’s Special Report on Global Warming of 1.5 °C, noting that the Report “confirms that climate change poses significant risks to the enjoyment of the human rights … [and that] … [t]he adverse impacts identified in the report, threaten, among others, the right to life, the right to adequate food, the right to adequate housing, the right to health, the right to water and cultural rights [14]”. The Joint Statement further notes that “[t]he risk of harm is particularly high for those segments of the population already marginalized or in vulnerable situations or that, due to discrimination and pre-existing inequalities, have limited access to decision-making or resources, such as women, children, persons with disabilities, indigenous peoples and persons living in rural areas [14]”. It also notes that “[c]hildren are particularly at heightened risk of harm to their health, due to the immaturity of their body systems [14]”. It concludes that

“State parties have obligations, including extra-territorial obligations, to respect, protect and fulfil all human rights of all peoples … [and that their] … [f]ailure to take measures to prevent foreseeable human rights harm caused by climate change, or to regulate activities contributing to such harm, could constitute a violation of States’ human rights obligations”.[14]

While the human rights practitioners have often trifurcated the obligations to “respect, protect and fulfil [15]”, in practice this distinction may have limitations [16] and, in any event, all three are part of a State’s responsibility to abide by its international legal obligations [6,17]. Thus, in the below discussion of States’ legal obligations and examples of policies and practices, we rely on a holistic understanding of States’ international legal obligations.

### 2.1. The Right to Health

The right to health is firmly established in international law [18,19,20,21,22,23,24,25,26,27,28,29,30,31]. It has generally been defined as “a state of complete physical, mental and social well-being and not merely the absence of disease or infirmity … [and] … [t]he enjoyment of the highest attainable standard of health is one of the fundamental rights of every human being…” [18,19,20,21,22,23,24,25,26,27,28,29,30,31]. A more specific definition of the right to health in Article 12(1) of the International Covenant of Economic, Social, and Cultural Rights (ICESCR) defines it as “the right of everyone to the enjoyment of the highest attainable standard of physical and mental health”, and paragraph 2 sets out a number of “steps to be taken by the State parties […] to achieve the full realisation of this right [18,19,20,21,22,23,24,25,26,27,28,29,30,31]”.

The Committee on Economic, Social and Cultural Rights (CESCR) points out that “the right to health embraces a wide range of socio-economic factors that promote conditions in which people can lead a healthy life, and extends to the underlying determinants of health, such as food and nutrition, housing, access to safe and potable water and adequate sanitation, safe and healthy working conditions, and a healthy environment [15]”. The CESCR also points out that “the right to health includes certain components which are legally enforceable [15]”. Thus, while the ICESCR provides for the progressive realization of the right to health, it also imposes obligations that are of immediate effect [32], such as the “obligation to move as expeditiously and effectively as possible towards the full realisation of Article 12 [15]”. These statements of legal obligation unequivocally evidence the intention of States to protect human health, including from climate change. 

### 2.2. States’ Obligations to Protect the Right to Health in the Context of Climate Change

In both climate change and human rights forums, States have recognized that they have the obligation “to undertake measures to mitigate climate change and prevent its negative impacts on human rights; to ensure all persons have the capacity and means to adapt; and to ensure accountability and an effective remedy for harms caused by climate change [6,33]”, especially harm to human health. Both human rights [33,34] and climate change [33,35] law also require preventive action by States to prevent or mitigate harm. These obligations include duties to provide adequate financing for climate action in developing states [5] and obligations arising from the precautionary principle to prevent loss and damage [5,34].

The UNFCCC requires all Parties, in accordance with their common, but differentiated, responsibilities and respective capabilities, to formulate and implement programs containing measures to mitigate the consequences of climate change. Mitigation measures can contribute to the reduction of vulnerabilities and risks, and decrease the likelihood of adverse health impacts [5]. The Paris Agreement adds to this framework a long-term temperature goal of keeping global warming “well below” 2 C above pre-industrial levels, and pursuing efforts to keep it below 1.5 C [6]. All parties are obliged to pursue domestic mitigation measures, with the aim of achieving the objective of their NDCs [6]. These efforts must be undertaken in accordance with the principles of progression and the highest possible ambition [6], while developed States remain bound by their obligations to provide financial resources to assist developing States with respect to both mitigation and adaptation [6]. States are obliged to undertake these legal obligations in good faith [36,37,38,39].

The UNFCCC also calls on all States to employ appropriate methods, such as impact assessments aimed at minimizing the adverse effects of measures taken to respond to climate change on, amongst other things, public health [5]. This entails measures to foster sustainable economic development at the local and regional levels [40], which increase, rather than undermine, the resilience and adaptive capacity of persons and groups of persons at risk of suffering adverse effects on health. Substantively, such responses may involve investing in measures that improve housing, livelihood diversification, education, food security, and health care [41], as well as establishing social protection, such as safety net schemes [42].

As noted above, mitigation and adaptation measures must be supported by climate financing. ‘Climate finance’ refers to local, national, or transnational financing that seeks to support mitigation and adaptation actions that will address climate change [43]. On 28 March 2022, at a preparatory meeting for the 50-year anniversary of the UN Conference on the Human Environment, the G77 + China focused their attention on financing for addressing climate change, stating that it must be a priority for any future development agenda [44]. This concern is consistent with the principle of “common but differentiated responsibilities and respective capabilities”; the UNFCCC obliges developed States to provide financial resources to assist developing States parties in implementing its objectives [43]. Moreover, the UNFCCC formulates a distinct obligation for developed States to assist developing States that are particularly vulnerable to the adverse effects of climate change, including effects on human health [5], in meeting costs of adaptation to those effects [5]. In line with these obligations, developed States must provide climate finance through a variety of actions, including supporting country-driven strategies, and taking into account the needs and priorities of developing States [44]. 

The right to health requires developed States to consider the specific needs of developing States related to public health, and to assist developing States in addressing those needs. Considering that the negative impacts of climate change are “disproportionately felt by [persons] in vulnerable situations, particularly those living in geographically vulnerable developing countries [12]”, scaled-up assistance must be provided, particularly to the populations of these States that are at the greatest risk of suffering adverse health impacts. 

In situations where loss and damage cannot be prevented, international cooperation and assistance are especially critical in protecting the right to health. The obligation of international cooperation to realize the right to health is recognized in international treaties and reinforced by the commitment to a global partnership for sustainable development in Sustainable Development Goal (SDG) 17 [45]. According to the former UN Special Rapporteur on the Right of Everyone to the Enjoyment of Physical and Mental Health, Paul Hunt, “[i]n practice, the realization of the right to the highest attainable standard of health is dependent upon international assistance and cooperation [46]”. All States have obligations to cooperate with each other to realize the right to health based on customary international law, the UN Charter [47], and human rights treaties [18,19,20,21,22,23,24,25,26,27,28,29,30,31,32]. Developed States in particular must assist developing States to “take the lead in combatting climate change and the adverse effects thereof” in accordance with the principle of common but differentiated responsibilities [5]. This cooperation is especially important for developing States that are more vulnerable [12] and face significantly higher risks from climate change [10,11].

## 3. Accountability and Reparations

International law provides that, when a State is injured, it is entitled to compensation [48,49,50]. A corresponding right is recognized by international human rights instruments [51,52,53] and tribunals. The successive UN mandate-holders have agreed that “[u]nder international human rights law, there is a solid legal framework establishing the right of victims to reparation for gross human rights violations [54]”.

### 3.1. United Nations Climate Change Forums for Seeking Redress for Interference with the Right to Health Caused by Man-Made Climate Change

Although the UNFCCC contains provisions for resolving disputes that could be used to hold States accountable for actions inconsistent with the treaty, these provisions have not been used [5]. Moreover, there was initially no dedicated mechanism to redress the harm caused by climate change under the UNFCCC. This changed with the consideration of “loss and damage” in negotiations at the urging of Small Island States. Vanuatu, on behalf of the Alliance of Small Island States, submitted a proposal for an insurance mechanism to facilitate compensation for States affected by climate change, particularly sea-level rise [55,56]. While this proposal was not clearly included in the UNFCCC [5,57], advocacy for the establishment of a compensation and rehabilitation fund continued [58,59]. It ultimately culminated in the establishment of the Warsaw International Mechanism for Loss and Damage associated with Climate Change Impacts (WIM) at COP19 held in Warsaw, Poland, in 2013 [60]. The WIM is mandated to prevent, minimize, and address loss and damage in developing countries that are particularly vulnerable to the adverse effects of climate change.

By considering loss and damage, States intended to include in negotiations the adverse effects of climate change that have not been prevented by mitigation or adaptation [61,62,63,64]. In other words, to include all unavoidable, as well as unavoided, harm resulting from climate change [65]. The term also includes the harm caused by climate change adaptation, i.e., expenses and unintended consequences, as well as non-monetary or non-economic losses to include loss of livelihood, loss of biodiversity, and loss of human health [66]. The UNFCCC Secretariat has recognized that “[h]uman health incorporates physical, mental and social well-being, and its non-economic value stems from its contribution to well-being [66]”. 

The WIM tries to address loss and damage by increasing the knowledge and understanding of comprehensive risk management approaches to address loss and damage associated with the adverse effects of climate change, including slow onset impacts; strengthening dialogue, coordination, coherence, and synergies among relevant stakeholders; and by enhancing action and support, including finance, technology, and capacity-building to address the loss and damage associated with the adverse effects of climate change [61,62,63,64]. The WIM also assesses the risk of loss and damage associated with the adverse effects of climate change involving vulnerable communities and populations in its assessment [67]. Finally, another function of the WIM is to increase action and support, including finance, technology, and capacity-building to address the loss and damage associated with the adverse effects of climate change. 

Action and support enabled or provided through the WIM have the potential to address a range of threats to the right to health, such as preventing irreversible losses of ecosystems that sustain communities, strengthening the capacity of developing States to provide financial and other forms of assistance to those impacted by climate change, and to develop and implement strategies to end current and prevent future harm. However, as noted above, loss and damage are already occurring at a scale bound to increase as temperatures continue to rise. Whether the belatedly started WIM can actually prevent or reduce the impact of climate change on the right to health is still to be seen. 

In 2015, Article 8 of the Paris Agreement to the UNFCCC brought loss and damage unambiguously under the UNFCCC umbrella of concern [6]. At COP25 in Madrid, Spain, the Santiago Network on Loss and Damage was established and this year at COP26, the Santiago Network was operationalized and received several pledges of funding. While the Santiago Network aims to help persons who have suffered harm from climate change, it is not intended to provide redress for damage or accountability. Its assistance is more of an advisory and voluntary nature. It will likely function as an incubator that brings donors and implementing partners together and disseminates research. What it adds to existing bodies, such as the WIM, is yet to be seen. 

### 3.2. United Nations Human Rights Forums for Seeking Redress for Interference with the Right to Health Caused by Man-Made Climate Change

Academics have suggested that States should ensure “effective remedies in administrative or judicial proceedings for climate harm or the threat or risk of such harm, including modes of compensation, monetary or otherwise [68]”. Indeed, States are obliged to ensure access to effective remedies for violations of the right to health, including those occurring in connection with climate change [33,51,52,53,69]. 

The law on remedies for human rights violations reflects the general law on State responsibility [70]. As internationally wrongful acts, human rights violations trigger the distinct obligation for the responsible State to make full reparation for the injury caused by the wrongful act [70,71]. Reparation can be made through restitution, compensation, satisfaction, or a combination of these [71]. Each of these remedies is aimed at making the victim ‘whole’ [72] and denouncing the wrong that interfered with her or his rights.

Restitution is the primary remedy for human rights violations, as it is for other violations of international law [71], and involves “the restoration of the prior situation, reparations of the consequences of the violation, and indemnification for patrimonial and non-patrimonial damages” [73]. For violations of the right to health, the victim of the violation will usually be in the best position to identify ways in which their health may be restored to the greatest possible extent. Listening to victims and inquiring into their actual needs is, therefore, an indispensable element of providing redress for violations of the right to health, including violations related to climate change. 

Compensation is an appropriate form of reparation in situations where restitution is materially impossible. It involves assigning a monetary value to mental health injuries, governed by the overarching principle that compensation must be ‘proportional to the gravity of the violation and the circumstances of each case’ [74]. Comparative law and economic theory and practice can be of assistance in this assessment [72], and precedents exist with respect to quantifying damages, such as personal injury, including impacts on mental health [71]. These precedents demonstrate that awarding compensation for the health impacts of climate change is neither impossible, nor necessarily inappropriate. 

It should be noted, however, that restitution and compensation alone may not suffice to remedy violations of the right to health. Satisfaction and other forms of symbolic reparation may need to complement these remedies in order to achieve the ultimate aim of restoring the victim’s rights. In addition, rehabilitation involving medical and psychological care, as well as legal and social services [74], will often be an important component of redress for violations of the right to health.

As accountability and reparations are established principles of international human rights law, human rights bodies may prove valuable for providing redress to victims whose human rights are interfered with due to climate change. Indeed, individuals whose human right to health has been violated by State action or inaction that contributes to climate change may have access to both political or judicial and/or quasi-judicial forums within the United Nations human rights system. 

### 3.3. Political Forums

The Human Rights Council (the Council) is the preeminent political forum for upholding the human right to health within the United Nations System. The Council has considered climate change on numerous occasions, but merely in a descriptive manner without apportioning responsibility or even urging redress for those whose right to health has suffered interference. The first thematic resolution of the Council on climate change and human rights addressed the relationship between climate change and the right to health [75]. This resolution resulted in a panel discussion [75,76] and a report of the Office of the UN High Commissioner for Human Rights (OHCHR) on the topic [12].

In addition to the Council’s Special Procedure, mandate holders have considered climate change’s impact on human rights, but health has not been prominent [76,77]. In September 2021, the Council created the mandate of the Special Rapporteur on the promotion and protection of human rights in the context of climate change, who will start work in 2022 [78]. While the Council contributes to drawing attention to the interference with human rights caused by climate change, as a political body consisting of States, it is unlikely to agree on the apportionment of responsibility in the absence of agreement on this issue in the UNFCCC process. The Universal Periodic Review (UPR) procedure, through which every UN Member State’s human rights record is reviewed in a political process of reflection, criticism, and recommendations, has also addressed climate change [79]. According to a privately maintained UPR database, over 300 recommendations or voluntary pledges have been made concerning climate change, but, of those, only six have mentioned the right to health [80]. While the recommendations have been generally viewed favourably, the UPR process does not aim to provide redress for violations, but to urge States to cooperate to end and prevent future interference with human rights. Its effectiveness in achieving those goals has yet to be seen [81].

Although the Council has a complaint procedure, it is obscure, lacks transparency, and has not dealt with claims relating to climate change to date [82]. The Council could create a reparations procedure, as has been suggested by some academics [83], but it has not done so. Similarly, UNFCCC COPs have considered mechanisms to allow individual claims for interference with the right to health, but have failed to agree on any action [84]. Finally, it is worth noting that the World Health Organization has taken few steps to engage in protecting individuals’ health from interference from climate change.

The lack of political action by States is inconsistent with the intentions expressed in international instruments providing for legal obligations to protect the right to health. Assuming States have articulated values that they are seeking to protect and realize through their actions, a heightened degree of political commitment may be required in the political forums in which they express their priorities. 

### 3.4. Complaint Procedures

The United Nations has been increasingly engaged by individuals suffering from the adverse effects of climate change. On 10 November 2021, as COP26 was coming to a close and the progress appeared woefully inadequate, a group of children petitioned the United Nations Secretary–General to declare a climate emergency [85]. On 21 October 2021, Environmental Justice Australia submitted a communication on behalf of five young people to the UN Special Rapporteurs on Human Rights and the Environment, the rights of Indigenous peoples, and the rights of persons with disabilities, claiming that the Australian government’s inadequate NDCs and inaction on climate change violated their human rights [86].

Litigation indeed provides an important avenue for individuals and communities to seek redress for harms caused by climate change, and enhance accountability on the part of those causing these harms [33]. Generally, strategic climate litigation against governments has sought increased mitigation ambition, enforcement of existing domestic legislation and policy, and consideration of climate change as part of environmental review and permitting [87,88]. Climate change litigation is currently developing on an ad hoc basis, primarily at the domestic level, but also increasingly at the regional and international levels [89]. Climate change cases have been brought to the Court of Justice of the European Union, the Inter-American Court on Human Rights, the Inter-American Commission on Human Rights, the UN Human Rights Committee (HRC), and the UN Committee on the Rights of the Child (CRC Committee) [87,88,90,91].

Recently, climate change litigation is said to have taken a ‘human rights turn’, with a growing number of cases employing rights-based arguments [92]. While few of these cases have directly addressed the health impact of climate displacement, there is a potential for courts to play a role in the enforcement of States’ obligations to protect the right to health in the climate change context. Domestic avenues can include class actions targeting major groups of emitters or holding public officials responsible for failures of due diligence [89]. 

On 4 May 2020, sixteen children filed a communication with the CRC Committee using the Optional Protocol that allowed individual communications and claimed that five countries that were all party to the Convention on the Rights of the Child had violated the petitioners’ human rights by their failure to take adequate action to address the adverse effects of climate change [93]. On 12 October 2020, the Committee found that it had jurisdiction *ratione materiae* over the communication, as the petitioners had demonstrated the link between their harm and the States’ action or inaction, but found the communications inadmissible on the formal grounds of failure to exhaust domestic remedies [94]. A current and former UN United Nations Special Rapporteur on the issue of human rights obligations relating to the enjoyment of a safe, clean, healthy, and sustainable environment filed an *amicus curiae* brief calling for the Committee to decide the communications in favour of the children [95]. The Special Rapporteurs concluded that the Committee should act because “[t]he time for action to address the climate crisis and prevent catastrophic impacts on children’s rights is rapidly running out [95]”.

Another potential avenue for redress for violations of the right to health connected with climate change consists of the complaint procedure under the ICESCR Optional Protocol [96]. This Protocol is currently ratified by twenty-four States [97]. Just like the respective mandates of the HRC and CRC, the CESCR’s mandate is limited to making recommendations to States. 

Although climate litigation is still in its infancy, these examples suggest that it has the potential to provide redress for violations of the right to health connected with climate change. Litigation can also be conducted for strategic reasons [98], such as raising awareness about climate change and its effects on health, pressuring States to enact or enforce domestic environmental and human rights legislation, and impacting treaty negotiations and State practice with respect to climate change and health.

However, litigation may not always be a suitable or appropriate avenue for redressing violations of the right to health associated with climate change, or it may simply not be available or accessible to victims. For instance, victims may not have access to courts because of legal, financial, linguistic, or other barriers. The contentious nature of litigation may also pose a psychological burden to victims and could deter some from seeking justice through the court system at all. Technical and procedural issues, such as jurisdiction, can also limit victims’ ability to have their case heard, with additional obstacles posed by potential judicial reluctance “to ‘develop’ the law on matters that are still subject to sensitive political negotiations [87,88]”. At a more fundamental level, domestic courts and quasi-judicial bodies are unlikely to be able to address the large-scale impacts of climate change on the right to health efficiently or at all. While international adjudication may be more capable of addressing the transnational and global dimensions of the health impacts of climate change, international adjudication is subject to its own limitations that may stand in the way of forging radical action on climate change or providing redress for victims [99].

It remains the case, however, that these legal and quasi-legal forums provide States with the ability to apply and appraise the application of the intentions they have expressed in legal instruments. It is noteworthy that States only voluntarily submit to these procedures and, therefore, must be understood as submitting to the appraisal of the values that they have articulated in the legal instruments they have created to guide their actions. 

## 4. Conclusions

Our brief observations on the articulation of policies and prescriptive legal obligations by States in human health and climate change evidence a widely held commitment to protecting human health from the adverse effects of climate change. This commitment has been repeatedly reiterated in the rhetoric of States. In particular, increasing efforts are now being made to use legal and aspirational instruments to address the health impact of climate change. These efforts take place at the multilateral level and through appeals to bodies that can interpret and apply the relevant instruments to ensure the right to health in the context of climate change. Yet, to date, few international bodies have made pronouncements that further protect the right to health in a manner that is commensurate with the urgency of the problem. Moreover, there remains a noticeable gap between States’ undertakings in international legal instruments, policies, and practices. This situation is concerning because climate change is still—despite a global pandemic—likely to be the greatest challenge to global health in this century, and indeed centuries to come. It is consistent with States’ expressions of intention that they now take action to achieve the values they have articulated.

One means of reversing the slide and moving the international community in the direction of achieving not only the letter, but also the spirit, of the legal commitments that have been taken in both climate change and human rights instruments is the creation of an effective implementation mechanism with clear jurisdiction *ratione materiae* over the rights of individuals and peoples that are interfered with or violated by the adverse effects of climate change. While proposals for an international environmental court or tribunal have floundered in the past, their revival could be an appropriate statement by States that they are serious about implementing their international legal obligations in this realm. At the same time, it is hoped that existing courts and human rights bodies will utilize their respective mandates to provide enhanced legal protection against preventable violations of the right to health connected with climate change, and to ensure access to redress for violations that have already occurred. The stakes have never been higher. 

## Data Availability

Not applicable.

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
