# Peer review of "Protecting Human Health from Climate Change: Legal Obligations and Avenues of Redress under International Law"

_ijerph, 2022, doi:10.3390/ijerph19095386_

Round 1
Reviewer 1 Report
The content of the abstract is the same as that of the introduction and should be modified appropriately. Additionally, there are incorrect time expressions such as 'this year at COP26'.
Part II of the paper highlights the obligations of states to protect the right to health. It proposes that both human rights and climate change law require states to take preventive action to prevent or mitigate harm. However, it seems a little fragmented, not systematic. The adverse effects of climate change on health can be classified, one is the adverse health effects of climate change, and the other is the adverse health effects of measures to combat with climate change. It could also be considered to categorize states' obligations to protect the right to health, such as the obligation to respect, protect, promote. Make this part more systematic and clearer. It will make this part more systematic and clearer.
The paper proposes several accountability and compensation mechanisms in Part III, but this part is too descriptive, with the absence of adequate and necessary, in-details analysis. It is possible to further analyze the reasons for the failure of current remedies. For instance, the causality of liability for climate change damage is difficult to determine. It is highly suggested to broaden the content in order to make analysis and conclusion more convincing.
Finally, several recommendations related to future development could be considered in the conclusion section, such as urging countries to meet their obligations, strengthening international cooperation in the field of health, and ensuring that mitigation and adaptation measures do not undermine the right to health.
Author Response
We are grateful to the reviewer for providing such helpful feedback. We have made extensive revisions to the manuscript to accommodate the concerns raised. Specifically, we have explained how both the adverse effects of climate change and measures to respond to climate change can have adverse health effects (with references to AR6 WGII). In addition, we added references to the respect-protect-fulfil typology. In the part on accountability and compensation mechanisms, we have sharpened the analysis. In the concluding section, we have added further thoughts on the creation of a new mechanism. We trust that the reviewer's concerns have been adequately addressed through these revisions, and wish to express our thanks again for the helpful feedback.
Reviewer 2 Report
This study explored how human health can be protected from climate change and its adverse effects by reliance on States’ obligations under international law. The topic is very important and worth to be published in IJERPH. The paper is well written and I therfore recommend it for publication in current form.
Author Response
We are grateful for these supportive comments.
Reviewer 3 Report
This is an excellent, "cutting edge" article, very different from most writing in international law from recent decades. I think that the contribution and differences need to be made explicit, particularly that the authors' view of international law (which I share) is not that of many governments such as Australia and the United States. Although a journal article is different from the daily news, something about shifts in the US Biden administration and on the role of climate activists would be useful. Rather than looking for a yes/no answer on whether the Committee on the Rights of the Child make international law something on presumptions of hardening of soft law would be useful. Put simply: It's a great article but the authors need to state up front in simple terms why, whose views will need to change, and why international law must consider what's above and what's within states.
Author Response
We are grateful to the reviewer for providing such helpful feedback. We have made extensive revisions to the manuscript to accommodate the concerns raised. Specifically, we have provided clarity about our perspective and integrated further analysis from the policy-oriented approach throughout the article. We also sharpened the final part of the article and added thoughts about the creation of a new mechanism. We trust that the reviewer's concerns have been adequately addressed through these revisions, and wish to express our thanks again for the helpful feedback.